# Effects of Flupyradifurone and Two Reference Insecticides Commonly Used in Toxicological Studies on the Larval Proteome of the Honey bee *Apis mellifera*

**DOI:** 10.3390/insects14010077

**Published:** 2023-01-12

**Authors:** Arne Kablau, Silvio Erler, Jakob H. Eckert, Jens Pistorius, Soroush Sharbati, Ralf Einspanier

**Affiliations:** 1Institute of Veterinary Biochemistry, Freie Universität Berlin, 14163 Berlin, Germany; 2LABOKLIN GmbH and Co. KG, 97688 Bad Kissingen, Germany; 3Institute for Bee Protection, Julius Kühn Institute (JKI)—Federal Research Centre for Cultivated Plants, 38104 Braunschweig, Germany; 4Zoological Institute, Technische Universität Braunschweig, 38106 Brauschweig, Germany; 5Institute for Microbiology, Technische Universität Braunschweig, 38106 Brauschweig, Germany

**Keywords:** sublethal effect, insecticide, larval development, pesticide, 2D protein electrophoresis, protein identification

## Abstract

**Simple Summary:**

Insecticides are considered to be one of the major factors of bee decline. In this study, the potential sublethal effects of selected neonicotinoids on honey bee larvae were investigated by protein expression profiling for the first time. The total larval protein expression was investigated by 2D gel electrophoresis after exposure to the insecticides dimethoate, fenoxycarb and flupyradifurone. Protein spots whose concentrations differed significantly from the controls were sequenced and identified against known insect proteins. Although the treated larvae did not show increased mortality or an aberrant development, the proteome comparisons showed differences in the metabolism, immune response and energy supply of the bee larvae. The strongest influence was found for flupyradifurone, which activates various detoxification pathways, the immune response or tissue regeneration. Our results suggest that there may be a delayed larval development or possibly a reduced honey bee brood vitality at sublethal concentrations.

**Abstract:**

The western honey bee *Apis mellifera* is globally distributed due to its beekeeping advantages and plays an important role in the global ecology and economy. In recent decades, several studies have raised concerns about bee decline. Discussed are multiple reasons such as increased pathogen pressure, malnutrition or pesticide use. Insecticides are considered to be one of the major factors. In 2013, the use of three neonicotinoids in the field was prohibited in the EU. Flupyradifurone was introduced as a potential successor; it has a comparable mode of action as the banned neonicotinoids. However, there is a limited number of studies on the effects of sublethal concentrations of flupyradifurone on honey bees. Particularly, the larval physiological response by means of protein expression has not yet been studied. Hence, the larval protein expression was investigated via 2D gel electrophoresis after following a standardised protocol to apply sublethal concentrations of the active substance (flupyradifurone 10 mg/kg diet) to larval food. The treated larvae did not show increased mortality or an aberrant development. Proteome comparisons showed clear differences concerning the larval metabolism, immune response and energy supply. Further field studies are needed to validate the in vitro results at a colony level.

## 1. Introduction

The western honey bee *Apis mellifera* is considered to be one of the world’s most important farm animals, mainly due to its high importance as a pollinator, pollinating more than 70% of global crops [1,2]. Within the last two decades, scientists observed not only a (local) decline of honey bee colonies, but also of bees and insects in general [3,4,5,6]. It is assumed that multiple factors contribute to this decrease such as the ectoparasitic mite *Varroa destructor* and accompanied virus infections, habitat loss, malnutrition and an increased application of insecticides as well as other plant protection products [7,8]. There are different classes of insecticides on the market that differ in their chemical structure and modes of action. Most widely used are the so-called neonicotinoids [9]. They belong to the group of competitive modulators of the nicotinic acetylcholine receptor (nAChR) and lead to the constant activation of cholinergic receptors, resulting in the death of insects [10]. In 2013, the European Food Safety Authority (EFSA) published a report summarising their assessment of the toxic effects of neonicotinoids on honey bees. As a consequence, the field use of three nitro-substituted neonicotinoids (clothianidin, thiamethoxam and imidacloprid) was banned in the EU [11]. Thus, in 2018, additional data were examined, including reports on the resistance [12], impact on wildlife [13,14] and toxic effects on wild bees [15], which confirmed the first assessment of 2013.

Flupyradifurone is a next-generation butenolide insecticide that belongs to the same group as neonicotinoids [16], acting as a nicotinic acetylcholine receptor (nAChR) competitive modulator. Its oral LD_50_ for honey bees is specified to be 1.2 µg of the active substance (a.s.)/bee [16]. Primarily, flupyradifurone may be ingested through nectar, pollen or by contact exposure during spray applications. In adult worker honey bees, exposure to flupyradifurone led to impaired motor abilities [17], affected olfactory learning [18] or altered the immune responses [19]. However, not only are adult worker bees essential for a healthy colony, but also the honey bee brood. Pesticides detected in pollen, nectar and combs led to chronic exposure of the larvae inside a hive [20,21,22]. To test the impact of insecticidal active compounds on honey bee broods, Aupinel and colleagues [23] established an in vitro larvae feeding test under standardised conditions following chronic exposure to chemical substances. The protocol monitored the mortality and larval development, and gave rise to the development of the OECD Guidance Document 239 [24]. Whilst sublethal concentrations may not lead to mortality, the sublethal effects should also be considered for an integrated risk assessment [25]. Therefore, we conducted a larval feeding assay following chronic exposure, as established by the OECD [24], to study the larval physiological responses by means of the protein expression after insecticide exposure. The focus was on the next-generation butenolide flupyradifurone; two commonly used reference insecticides (dimethoate and fenoxycarb) were used as the reference items, as utilised in standard honey bee laboratory assays, semi-field assays and field assays for regulatory purposes. The inclusion of dimethoate and fenoxycarb allowed us to consider the further insecticidal modes of action in the study. However, based on the results of this study, it was not possible to provide a defined set of candidate proteins to detect the potential adverse effects of sublethal concentrations of xenobiotic substances.

## 2. Materials and Methods

### 2.1. Honey Bees

In vitro experiments were performed during spring and summer 2018/19 using three queen-right colonies (each colony was equal to one replicate) with *Apis mellifera* (Buckfast) sister queens. The colonies remained at the Institute for Bee Protection at the Julius Kühn Institute in Braunschweig (Braunschweig, Germany). The experimental colonies were healthy, had a sufficient food supply and showed no symptoms of disease or increased parasitism. No medical treatments (e.g., varroacides) were given four months before running the experiments.

### 2.2. Honey Bee Brood Test

Honey bee larvae (*n* = 16 per colony; *n* = 48 per insecticide treatment and controls) were reared and exposed according to the protocols of the OECD Guidance Document 239 [24] and Kablau et al. [26]. The larvae were fed individually with different diets using organic royal jelly (Cum Natura) that was analysed for contaminants and contained no residues of xenobiotics. Pilot experiments were performed to determine the sublethal concentrations of dimethoate and fenoxycarb for the experimental setup (data not shown). The sublethal concentration of flupyradifurone was selected according to EFSA [27], with a no-observed-effect concentration (NOEC) of ≥ 10 mg a.s./kg diet.

Flupyradifurone (HPC Standards, purity 99.9%), dimethoate (BASF, purity 99.9%) and fenoxycarb (HPC Standards, purity 98.3%) were not used in combination with other substances. Acetone was used to prepare the stock solution of flupyradifurone and all subsequent dilutions. The test solution in the final diet was 0.5% *w/w*. The diet of the solvent control group contained 0.5% acetone to preclude the effects of the solvent itself. The larvae were fed from day three (D3) until day six (D6) post-grafting, with a constant concentration of flupyradifurone that was dissolved in the diet. This resulted in a cumulative dose on D6, according to Table 1. The organophosphate dimethoate (AChE inhibitor; water as a solvent and content in the final diet of 10% *w/w*) and the carbamate fenoxycarb (insect growth regulator; acetone as the solvent and content in the final diet of 0.5% *w/w*) were used as additional test substances, with a known toxicity in developing honey bee larvae. Previous experiments using sublethal concentrations showed no increased mortality between a water control and an acetone control, and no differences in mortality or development between the control and treatment groups. Based on the results of a previous transcriptome study [26], the living larvae were collected on day 6 (dimethoate and fenoxycarb) and 8 (flupyradifurone) (4 individuals per insecticide treatment group and the control), washed with PBS, snap-frozen in liquid nitrogen and stored at −80 °C until further processing.

### 2.3. Protein Analysis

#### 2.3.1. Preparation of Protein Extracts and Two-Dimensional PAGE (2D-PAGE)

Each honey bee larva was homogenised in a 300 µL lysis buffer (9 M urea; 2% CHAPS) using bead tubes (Lysing Matrix D, Mpbio, Heidelberg, Germany) and a bead-based homogeniser (Bead Blaster 24^TM^, Benchmark). Amounts of 10 mg DTT, 5 µL PMSF and 1 µL protease inhibitor cocktail (Merck Biosciences) were added to a 1 mL lysis buffer before the homogenisation. After incubation on ice for 1 h, the samples were centrifuged (10 min; 13,000 × *g*; 4 °C) and the supernatant was used for a further clean-up following the protocol of the manufacturers (2D Clean Up Kit, GE Healthcare, Chicago, IL, USA). Finally, the dried protein samples were dissolved in an appropriate volume of a DIGE buffer (8 M urea; 4% CAPS; 30 mM Tris, pH 8.5). The total protein concentration was measured in duplicate using a Pierce 660 nm Protein Assay (Thermo Scientific, Waltham, MA, USA) and bovine serum albumin as a standard (50 µg/mL–2 mg/mL).

For the proteome analysis, single protein samples were labelled with either 400 pmol Cy3 or Cy5 (CyDyes, GE Healthcare), respectively. Cy2 was used as an internal standard (common reference) containing equal amounts of the pooled samples. Next, 2D gels were loaded with 50 µg total protein of a mixture of each Cy3-, Cy5- and Cy2-labelled sample in a 300 µL volume of DeStreak^TM^ Rehydration solution (GE Healthcare). Isoelectric focusing (IEF) was performed within a pH range of 3 to 10 using immobilised pH gradient (IPG) strips (ReadyStrip, BioRad) and a Protean IEF Cell (BioRad) system at 20 °C with the following cycle: 50 V for 14 h, 200 V for 1 h, 500 V for 1 h, 10,000 V for 1 h and 10,000 V for 6 h.

For the molecular weight analysis, the IPG strips were incubated for 15 min in an equilibration buffer (6 M urea; 30% glycerol; 2% SDS; 50 mM Tris, 0.02% bromophenol blue, pH 8.8) containing 10 mg/mL DTT and equilibrated for 15 min with an equilibration buffer containing 25 mg/mL iodoacetamide. The strips were transferred onto vertical 12.5% SDS-PAGE gels and sealed with 0.5% low-melting-point agarose. The second-dimension separation was run using an Ettan DALTsix electrophoresis unit (GE Healthcare) with the following parameters for six gels: 2 W for 1 h and 12 W. This continued until the dye front reached the end of the gels.

Protein spots were visualised with a Typhoon 9400 fluorescence scanner (Amersham Biosciences, Amersham, UK) at the respective wavelengths of the Cy dyes. The spot detection and the matching and quantification of the spot intensity were performed using DECODON Delta 2D software (version 4.5.3, Greifswald, Germany). Only spots with more than a 1.5-fold difference in density (enhanced or decreased expression) were considered for the subsequent protein analysis. All gels and protein analyses were performed with four biological replicates.

#### 2.3.2. Protein Picking and Identification

A second 2D gel electrophoresis was run with 400 µg protein per gel to pick the proteins with a significant fold change between the treatment groups. Multiple lysates per treatment group (*n* = 4) were mixed and filled with DeStreak^TM^ Rehydration Solution to a final volume of 300 µL in the absence of Cy dyes. A protein ladder (Spectra^TM^ Multicolor Broad Range Protein Ladder, Thermo Scientific) was used to estimate the molecular weights of the target proteins. The gels were silver-stained and the selected spots (see Section 2.3.1) were manually excised from the gel and stored in a buffer (30 mM Tris; 8 M urea; 4% CHAPS). The protein identification (LC-MS) was achieved by Proteome Factory Berlin (Proteome Factory AG) (a detailed description of the methods can be found in the Appendix A). Trypsin was used for the protein digestion. For the identification of the proteins, MASCOT Search Engine (Matrix Science) and sequences of *Apis mellifera* were used. As fixed (f) and variable (v) modifications, the following were chosen: carbamidomethyl (f) of cysteines; oxidation (v) of methionine; and deamidated (v). The matched peptides of the identified proteins after 2D gel electrophoresis and the identification via LC/MS are summarised in Appendix A.

### 2.4. Statistical Analysis

The image analysis and spot quantification of the protein spots were performed with DECODON Delta2D 4.5.3. software (DECODON GmbH). Differences in the expression between the samples (a minimum of a 1.5-fold change between the treatment vs. the solvent control) were analysed using unpaired two-tailed Welsh’s t-tests (*p* < 0.05).

## 3. Results

A protein analysis of 8-day-old larvae treated with sublethal concentrations of flupyradifurone resulted in 951 detectable protein spots in total. Of these, 129 spots showed differential intensities, with 22 upregulated and 107 downregulated proteins. Peptide mapping (LC/MS) identified five selected protein spots (Table 2, Figure 1). Two of the proteins that responded to the flupyradifurone treatment were enzymes of metabolic pathways: retinal dehydrogenase 1 (XP_392104.4; mean fold difference to the control: +1.78; *p* < 0.001) and 3-ketoacyl-CoA thiolase (XP_391843.1; mean fold difference to the control: −15.96; *p* = 0.009). In addition, major royal jelly protein 2 (MRJP2, NP_001011580.1; mean fold difference to the control: −30.99; *p* < 0.001) and 14-3-3 protein zeta (XP_006566156.1; mean fold difference to the control: −1.6; *p* = 0.01) were significantly downregulated.

For the two reference insecticides, a lower number of differentially regulated spots was detected (dimethoate: 45 upregulated and 26 downregulated; fenoxycarb: 6 upregulated and 13 downregulated) after treating the larvae with sublethal concentrations (Figure 1). Three of the identified proteins of the dimethoate group were retinal dehydrogenase 1 (mean fold difference to the control: −1.60; *p* = 0.02), 3-ketoacyl-CoA thiolase (mean fold difference to the control: −1.68; *p* = 0.02) and glutathione S-transferase S1-like protein (mean fold difference to the control: −1.57; *p* < 0.001). Major royal jelly protein 2 was again downregulated (mean fold difference to the control: −2.11; *p* = 0.02) (Table 2). Within the fenoxycarb treatment, only MRJP2 was significantly downregulated (mean fold difference to the control: −4.10; *p* < 0.001) (Figure 1; Table 2).

## 4. Discussion

The chronic exposure of honey bee larvae to sublethal concentrations of dimethoate, fenoxycarb and flupyradifurone caused significant changes in the protein regulation (mostly downregulation) without increasing the mortality or affecting the larval development. Five proteins were identified as significantly differentially regulated; major royal jelly protein 2 was downregulated in all treatment groups. Here, we only discuss the flupyradifurone-induced larval protein changes. However, it has to be mentioned that glutathione S-transferase S1-like protein was also downregulated in imidacloprid-treated honey bees [28]. This detoxification enzyme may play a more central role in the intoxication response and might be a candidate marker protein.

Insecticides lead to oxidative stress in many organisms and, consequently, to cell or tissue damage [29]. Natural antioxidants such as carotenoids and polyphenols in pollen serve as protective agents against free oxygen radicals [30] and are consumed by adult bees and larvae via their diet. In addition, carotenoids serve as precursors for retinoids. The conversion of retinaldehyde to retinoic acid is mediated by the enzyme retinal dehydrogenase 1 [31]. In insects, retinoids are involved in processes ranging from vision to morphogenesis [32,33]. Halme et al. [34] showed that flies have a retinoid-dependent pathway that delays pupation by inhibiting the ecdysone expression. This leads to a delayed larval development and could promote tissue regeneration, as was found in vertebrates [35]. The induced expression of a central enzyme of the retinoid pathway might support this observation. Neonicotinoid (imidacloprid) intoxication also induces retinal dehydrogenase 1 upregulation [28]. Flupyradifurone might cause tissue damage (as shown by increased apoptosis for the commercial product Sivanto^TM^ feed to adult honey bees [36]) and may lead to an enhanced retinoid metabolism to support tissue regeneration. In the case of dimethoate, a decreased expression may accelerate the preparation for pupation, reducing the time for potential tissue damage.

3-Ketoacyl-CoA thiolase was downregulated in all cases of insecticide treatments in the current and an earlier study [28]. This enzyme catalyses the final step of fatty acid β-oxidation in mitochondria [37]. A decreased expression might indicate a lower synthesis of hexamerins that function as larval storage proteins and as a source of amino acids [38]. Storage proteins provide essential nutrients during the aphagous pupal stage. The decreased breakdown of lipids might lead to an impaired xenobiotic response as intermediates of β-oxidation are also used as precursors in other cellular processes [39]; e.g., the immune system [40] or detoxification [41]. Furthermore, the intermediates of fatty acid metabolism are used for ATP production in the electron transport chain [42]. A previous gene expression study showed an impaired mitochondrial membrane transport after sublethal larval feeding with flupyradifurone [26], supporting this assumption. The suppression of 3-ketoacyl-CoA thiolase could negatively affect the energy supply via ATP production as well as the supply of intermediates for other metabolic pathways.

Sublethal doses of flupyradifurone significantly reduced the 14-3-3 protein zeta abundance although this negative trend was also found for the other two insecticides. The family of 14-3-3 proteins are highly conserved in eukaryotes, ranging from yeast to mammals [43,44,45]. In insects, 14-3-3 proteins are involved in survival [46], neuronal differentiation [47,48], cell proliferation and cell death [49] as well as organ development [50]. Phagocytosis, which represents a primary response of the innate immune mechanism of insects, also appears to be regulated by 14-3-3 proteins [51,52]. The observed downregulation of 14-3-3 protein zeta could, therefore, affect the larval development or immune response. This is very consistent with previous transcriptome data that showed a significant downregulation of 14-3-3 protein zeta in flupyradifurone-treated honey bee larvae [26] and with previous proteomic studies in honey bees treated with fipronil, imidacloprid and pyraclostrobin [28,53].

Major royal jelly protein 2 was strongly downregulated after feeding the larvae with sublethal concentrations of all three insecticides. Comparative observations have been described for pyraclostrobin (fungicide)- and fipronil (insecticide)-fed nurse bees [53]. Royal jelly is secreted by young worker honey bees to feed the worker larvae during the first three days [54]. Primarily, major royal jelly proteins serve as nutrients for honey bee broods and the queen [55,56]. Nevertheless, in the last years, more functions of this protein family have been discovered, including the exchange of RNA between workers and larvae as a possible driver for social immunity [57] or antimicrobial and antioxidant activities [55,58]. At this stage, however, it is not yet clear what other functions MRJPs may play in addition to fulfilling the nutritional needs of bees. The reduced quantity of MRJP2 might be compensated by an increased feeding behaviour.

In summary, this protein analysis showed that dimethoate, fenoxycarb and flupyradifurone had an impact on larval metabolism and development, but without any clear pattern related to their specific mode of action. Flupyradifurone had the strongest impact on the protein change among the list of identified protein spots. The activation of different pathways initiating detoxification, the immune response or tissue regeneration in honey bee larvae might increase their metabolic rate and, therefore, the need for nutrients that are provided by carbohydrates and major royal jelly proteins [59]. An alternative explanation might be that the consumption of xenobiotics leads to a decreased food intake caused by the substance itself or indirectly by activating other pathways, resulting in an inhibition of larval growth and development [60,61,62]. Both possibilities might finally result in a retarded larval development or, in the worst case, the death of the honey bee brood. To detect any further effects, it could be useful to determine the weight of the larvae or adults, as recommended by the US Environmental Protection Agency [63].

## 5. Conclusions

Following the OECD Guidance Document 239 on honey bee larval toxicity tests, this study showed differences in protein abundance with regard to larval metabolism, the immune response and energy supply after feeding honey bee larvae with sublethal concentrations of three different insecticides. During the larval development, the larvae showed relatively strong effects on the protein expression, which was consistent with previous larval transcriptome data. The sublethal concentrations of the tested insecticidal substances did not cause an increase in mortality or affect the larval development [26]. Regarding a holistic risk assessment, field trials are required to test the effects at a colony level and in realistic exposure scenarios in the field. With the exception of major royal jelly protein 2, it is difficult at this stage to recommend a set of candidate proteins for the identification of the potential adverse effects of sublethal concentrations of xenobiotic substances. This study showed that there was a strong effect on the protein expression, but further work is needed to assess the impact of realistic exposure scenarios on individual bee health as well as their longevity and ability to function normally as a part of a colony.

## Figures and Tables

**Figure 1 insects-14-00077-f001:**
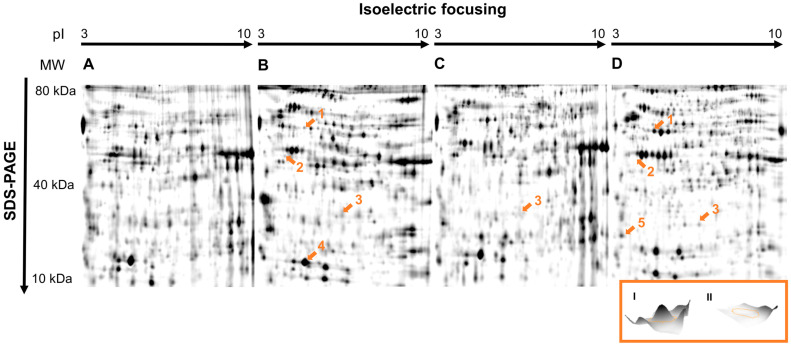
The 2D proteome maps of honey bee larvae exposed to (**A**) solvent control (CS), (**B**) dimethoate (T1), (**C**) fenoxycarb (T2) and (**D**) flupyradifurone (T3). Dysregulated and identified spots are marked with orange numbers and arrows (1: retinal dehydrogenase 1; 2: 3-ketoacyl-CoA thiolase; 3: major royal jelly protein 2; 4: glutathione S-transferase S1-like protein; 5: 14-3-3 protein zeta). MW: molecular weight in kDa; pI: isoelectric point. Inlay (I, II): the 3D profile of 3-ketoacyl-CoA thiolase in control larvae (I) and after exposure to flupyradifurone (II).

**Table 1 insects-14-00077-t001:** List of sample groups, corresponding test substances and their respective concentrations and cumulative doses.

Group	Treatment	Concentration (mg a.s./kg Diet)	Cumulative Doses (µg a.s./Larva)	Active Substance
C	Control	-	-	Water
CS	Solvent control	-	-	Water + acetone
T1	Insecticide	1.29	0.2	Dimethoate
T2	Insecticide	0.32	0.05	Fenoxycarb
T3	Insecticide	10	1.54	Flupyradifurone

a.s.—active substance.

**Table 2 insects-14-00077-t002:** All identified significantly regulated proteins of the three different insecticide treatment groups. Significant changes in relation to their respective solvent controls (CS) are marked in bold.

Protein	Dimethoate (Mean Fold Difference)	*p*-Value (*t*-Test)	Fenoxycarb (Mean Fold Difference)	*p*-Value (*t*-Test)	Flupyradifurone (Mean Fold Difference)	*p*-Value (*t*-Test)	No. of Matched Peptides
**Glutathione S-transferase S1-like**	−1.57	**0.0005**	−1.02	0.86	1.17	0.37	633
**3-Ketoacyl-CoA thiolase**	−1.68	**0.02**	−1.42	0.20	−15.96	**0.009**	58
**Major royal jelly protein 2**	−2.11	**0.02**	−4.10	**0.0002**	−30.99	**0.0007**	43
**14-3-3 Protein zeta**	−1.17	0.35	−1.21	0.44	−1.60	**0.01**	66
**Retinal dehydrogenase 1**	−1.60	**0.02**	1.16	0.43	1.78	**0.0002**	120

## Data Availability

Data supporting the reported results are available from the corresponding author upon reasonable request.

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
