# Peer review of "Effects of Flupyradifurone and Two Reference Insecticides Commonly Used in Toxicological Studies on the Larval Proteome of the Honey bee Apis mellifera"

_insects, 2023, doi:10.3390/insects14010077_

Round 1

Reviewer 1 Report

Line 28 and 56 inaccurate claims that 3 neonicotinoids have been banned. This just applies for the EU, and for the field use. Moreover, the EU member states allowed several emergency approvals.

The method description does not contain a clear definition what a "x-fold difference" means. I would prefer a formula.

Section 2.4 is extremely short and just mentions "Student's t-test". My guess is that the authors used a two-sided test for independent samples. Distributions have probably not been checked for appropriateness (normal distribution, homoscedasticity). Maybe, this might not possible, due to low number of observations. But such should be mentioned.

Are there missing values? If so, how has this problem been solved?

Table 2 should explicitly describe col 2, 4 and 6 headers as "mean fold-difference", and col 3, 5 aand 7 headers as "t-Test p".

Section 3 is hard to read and follow, being a duplication of table 2 content. It should be reworked to improve readability.

Author Response

Line 28 and 56 inaccurate claims that 3 neonicotinoids have been banned. This just applies for the EU, and for the field use. Moreover, the EU member states allowed several emergency approvals.

Answer: We agree with the reviewer and added the detail that these substances have been banned in the EU. In L54 of the original manuscript is was already mentioned that the substances were banned for field use.

Text has been changed to:

L28: …was prohibited in the EU.

L56: … in the EU.

The method description does not contain a clear definition what a "x-fold difference" means. I would prefer a formula.

Answer: We apologize for missing this part. Expression changes were estimated as relative values of the protein spot abundance of the treatment groups relative to the solvent control. By default, the ratio of spots in Delta 2D is shown as quotient of the two corresponding spots: if the relative volume of the second spot (control) is doubled compared to the first spot (treatment), the ratio is 2, if its volume is halved, the ratio is 0.5. A conventional ratio of 1:2 results in a fold change of -2, meaning a two-fold decrease. For the opposite 2:1 ratio the fold change is 2, meaning a two-fold increase. We have added this detail to Material and Methods.

L172: Differences in expression between samples (a minimum of 2-fold change between treatment vs. solvent control) were analysed …

Section 2.4 is extremely short and just mentions "Student's t-test". My guess is that the authors used a two-sided test for independent samples. Distributions have probably not been checked for appropriateness (normal distribution, homoscedasticity). Maybe, this might not possible, due to low number of observations. But such should be mentioned.

Answer: We did not check for normal distribution and homoscedasticity. Due to the low number of replicates we assumed a t-distribution and used an unpaired two-tailed Welsh t-test (unequal variances).

Are there missing values? If so, how has this problem been solved?

Answer: No

Table 2 should explicitly describe col 2, 4 and 6 headers as "mean fold-difference", and col 3, 5 aand 7 headers as "t-Test p".

Answer: We agree with the reviewer and have updated the column headings of Table 2 accordingly.

Section 3 is hard to read and follow, being a duplication of table 2 content. It should be reworked to improve readability.

Answer: Section 3 is a detailed description of the results of the study, referring to the results shown in Table 2 and Figure 1. In addition, this section also includes the general results on total number of spots identified and the number of  up- and down-regulated spots, which was not included in Table 2. Therefore, we would like to keep the current format and disagree with the reviewer in this case.

Reviewer 2 Report

As a novel insecticide that is thought to be relatively ‘bee safe’, flupyradifurone has been registered for use globally. However, flupyradifurone shares the same mode of action as neonicotinoids, raising the question as to whether it has similar sub-lethal impacts on beneficial insects such as bees. This manuscript provides the larval proteomic information of bees under the exposure to flupyradifurone and two reference insecticides, which is useful for a comprehensive risk assessment of flupyradifurone on bees. I have some suggestions as below for the improvement of this manuscript.

“Results”:

1.       The result content and the corresponding data seems a little simple. The protein analysis was preformed with four biological replicates, but only one gel image of each treatment was showed in Figure 1. It is better to provide all of the gel images including four repeats as supplementary data.

2.       Please provide the values of spot quantification of protein spots differentially expressed and their statistical analysis results as supplementary data. And add these values of the five identified proteins into Table 2, or draw a new image with 3D profile of of the five identified proteins in control larvae and after exposition to flupyradifurone, with all four repeats.

“Discussion”

3.       Please discuss more about the potential risk of flupyradifurone on bees, based on the present results and published references, which will be more interesting to the readers.

Author Response

As a novel insecticide that is thought to be relatively ‘bee safe’, flupyradifurone has been registered for use globally. However, flupyradifurone shares the same mode of action as neonicotinoids, raising the question as to whether it has similar sub-lethal impacts on beneficial insects such as bees. This manuscript provides the larval proteomic information of bees under the exposure to flupyradifurone and two reference insecticides, which is useful for a comprehensive risk assessment of flupyradifurone on bees. I have some suggestions as below for the improvement of this manuscript.

“Results”: 

  1. The result content and the corresponding data seems a little simple. The protein analysis was preformed with four biological replicates, but only one gel image of each treatment was showed in Figure 1. It is better to provide all of the gel images including four repeats as supplementary data.

Answer: Figure 1 shows an example of the protein maps for the respective compound exposure groups to illustrate the fold-change and support the data shown in Table 2. In addition, we now show all gels in the supplementary material.

  1. Please provide the values of spot quantification of protein spots differentially expressed and their statistical analysis results as supplementary data. And add these values of the five identified proteins into Table 2, or draw a new image with 3D profile of of the five identified proteins in control larvae and after exposition to flupyradifurone, with all four repeats.

Answer: The required data are additionally listed in a separate table in the supplementary material

“Discussion”

  1. Please discuss more about the potential risk of flupyradifurone on bees, based on the present results and published references, which will be more interesting to the readers.

Answer: Reading the discussion, it is clear that we already discussed potential effects on honey bees in relation to the specific proteins identified here. The study is an artificial in vitro approach, so we do not wish to extend the discussion further, as the results observed here may differ from field studies due to the design and setting. The observed effects of flupyradifurone on honey bees are mentioned at the introduction. The combination of our proteome results with the results of other studies might fail, as we investigated other specific parameters in our study. We may find a contexts that does not allow correlations with previously published data for this reason. Therefore, we decided, after some language improvements, to leave the discussion as it is to not overestimate our findings. Future studies should use field exposure experiments and comparative proteomic tools to validate the link between sublethal observations and proteomic results.

Reviewer 3 Report

This manuscript concerns an important topic and contains valuable information on the impact of pesticides on bee larval development. However, the methodology raises several concerns, especially pertaining to the choice of controls and the statistical analysis.

The authors justify the importance of this study based on the ban of three previously marketed neonicotinoids: “Flupyradifurone is dealt as a potential successor and has a comparable mode of action as the banned neonicotinoids”. I agree with this statement, however, one of the banned pesticides of the same family should have been included for comparison (positive control). This would have been a lot more important than including compounds with different modes of action.

The results of the range-finding experiments should be provided to appropriately justify the choice of dosage for each pesticide.

Why were larvae from dimethoate and fenoxycarb groups collected earlier (day 6) than those from the flupyradifurone group (day 8)? Many of the differences observed between these groups could simply be due to differences in larval development stages.

For consistency, please indicate in the text the concentration of each compound in the diet; I am uncertain why it was included for dimethoate and fenoxycarb (L109-110), but not for flupyradifurone (L106).

The method of euthanasia of the bee larvae is not specified.

Please provide the detailed LC/MS method (or reference if it has been previously published). The LC and MS conditions, including solvent, gradient, mode, etc. have to be specified. This could be added to the supplemental materials.

The statistical analysis section is incomplete. Student’s t-test is a parametric test. Data should have bee tested for normality of distribution first. Please verify data distribution, specify which test was used to do so, whether data was transformed (if so, how) to achieve a normal distribution and if non-parametric tests were used for data that could not be normalized. Also state if corrections were made for multiple comparisons.

From the methods and results, it is unclear how/why only 5 spots out of 129 were identified. This seems very low. It is unclear if identification was attempted for 129 and failed for all but 5, or if only some were selected for further identification, and if so, how selection was made. The same goes for the other two insecticides (only 3 spots identified). The results seem very limited for a full article (only 8 identified proteins?).

There are some awkward sentence structures and missing punctuation marks; English editing would improve the overall readability of this document.

Other minor comments

L14 : here for the first time

L19: remove However

L22: that, at sublethal concentrations, there may be…

L25: In recent decades, several…

L27: remove Especially

Author Response

This manuscript concerns an important topic and contains valuable information on the impact of pesticides on bee larval development. However, the methodology raises several concerns, especially pertaining to the choice of controls and the statistical analysis.

The authors justify the importance of this study based on the ban of three previously marketed neonicotinoids: “Flupyradifurone is dealt as a potential successor and has a comparable mode of action as the banned neonicotinoids”. I agree with this statement, however, one of the banned pesticides of the same family should have been included for comparison (positive control). This would have been a lot more important than including compounds with different modes of action.

Answer: Based on the aim of the study to provide a set of candidate proteins to detect possible adverse effects of sublethal concentrations of xenobiotic substances on honey bee larvae, the effects of the tested insecticides on the proteome were determined by comparison with larvae of a negative (untreated) control group. We agree with the reviewer that a positive control (e.g. use of a banned neonic) would have been an option for comparison in terms of mode of action, but this was not the aim of the current study. However, we intend to consider this point in future studies.

The results of the range-finding experiments should be provided to appropriately justify the choice of dosage for each pesticide.

Answer: The range-finding assay was conducted as a preliminary dose-response test to determine appropriate sublethal test concentrations. The highest test concentration at with there was no increase in larval mortality (when compared to the respective control group) was identified and included in the main study. Thus, the results would not add any significant information to this manuscript. It has already been mentioned that the test concentrations were determined in range-finding experiments and that chronic exposure of honey bee larvae to the tested sublethal concentrations neither increased larval mortality nor affected larval development (first sentence of the discussion). Therefore, we decided not to add the results of the range-finding experiments.

Why were larvae from dimethoate and fenoxycarb groups collected earlier (day 6) than those from the flupyradifurone group (day 8)? Many of the differences observed between these groups could simply be due to differences in larval development stages.

Answer: Regarding the laboratory feeding tests on honey bee larvae, the manuscript includes the results of two different tests conducted in 2019 and 2020. Due to divergent sampling schedules, larvae from the 2019 study were sampled on day 6 and larvae from the 2020 study were sampled on day 8. However, this variance should not affect the outcome of the study, as in this study the results of each insecticide are assessed independently. Furthermore, all exposed larvae were only compared with larvae of the same age in the respective control group. The results and discussion are based only on the comparison of treatment vs. control and not on the comparison of the different treatment groups with each other.

For consistency, please indicate in the text the concentration of each compound in the diet; I am uncertain why it was included for dimethoate and fenoxycarb (L109-110), but not for flupyradifurone (L106). 

Answer: The content of the test solution in the larval diet for flupyradifurone has already been reported (L104). Further details on test concentrations and doses per larva are given in Table 1, referenced in Material and Methods.

The method of euthanasia of the bee larvae is not specified.

Answer: Larvae were euthanized by submersion in liquid nitrogen. We update the text accordingly. L116: …, quick frozen in liquid nitrogen and stored …

Please provide the detailed LC/MS method (or reference if it has been previously published). The LC and MS conditions, including solvent, gradient, mode, etc. have to be specified. This could be added to the supplemental materials.

Answer: We have added the LC-MS-methodology used in supplementary material as follows:

Protein identification was performed by Proteome Factory (Proteome Factory AG, Berlin, Germany). Protein bands were rebuffered by repeated shrinking (60% acetonitrile, 50 mM TEAB) and swelling steps (50 mM TEAB). For Cysteine alkylation, the swelling buffer first contained 10 mM TCEP, then 10 mM 2-iodoacetamide. In-gel proteolysis was done with sequencing grade porcine trypsin (Promega, Mannheim, Germany) overnight. For analysis, the samples were acidified with 2% formic acid. The nanoHPLC-ESI-MS/MS system consisted of an Dionex Ultimate 3000 nanoHPLC system (Thermo Scientific, Germering, Germany), nanoelectrospray emitter (Fossiliontech, Madrid, Spain) and an Orbitrap Velos mass spectrometer (Thermo Scientific, Bremen, Germany). Peptides were first trapped and desalted on the enrichment column (PepMap-C18, 0.3 x 5 mm, Dionex) for five minutes (solvent: 0.5% acetonitrile/0.5% formic acid), then separated on a ReproSil-Pur 120 C18-AQ column (0.075 x 500 mm column, Dr. Maisch, Ammerbuch-Entringen, Germany) using a linear gradient from 12% to 40% B (solvent A: water, solvent B: acetonitrile, both with 0.1% formic acid) with a total run time of 35 minutes. Ions of interest were data-dependently subjected to MS/MS according to the expected charge state distribution of peptide ions. Proteins were identified by database search against the Apis mellifera subset (23491 sequences) of the RefSeq protein database (National Center for Biotechnology Information, Bethesda, USA) and a contaminant database using MS/MS ion search of the Mascot search engine (Matrix Science, London, England). Only peptides matches with a score of 20 or above were accepted. Protein matches were required to have at least two significant unique sequences.

The statistical analysis section is incomplete. Student’s t-test is a parametric test. Data should have bee tested for normality of distribution first. Please verify data distribution, specify which test was used to do so, whether data was transformed (if so, how) to achieve a normal distribution and if non-parametric tests were used for data that could not be normalized. Also state if corrections were made for multiple comparisons.

Answer: We did not check for normal distribution and homoscedasticity. Due to the low number of replicates we assumed a t-distribution and used an unpaired two-tailed Welsh t-test (unequal variances).

From the methods and results, it is unclear how/why only 5 spots out of 129 were identified. This seems very low. It is unclear if identification was attempted for 129 and failed for all but 5, or if only some were selected for further identification, and if so, how selection was made. The same goes for the other two insecticides (only 3 spots identified). The results seem very limited for a full article (only 8 identified proteins?).

Answer: The proteins of interest were selected according to their degree of expression change, repeatability in different gels, their minimum concentration and the possibility to isolate them from the gel without being contaminated by neighbouring protein spots.

There are some awkward sentence structures and missing punctuation marks; English editing would improve the overall readability of this document.

Answer: A native speaker has again checked and corrected the manuscript

Other minor comments

L14 : here for the first time

L19: remove However

L22: that, at sublethal concentrations, there may be…

L25: In recent decades, several…

L27: remove Especially

Answer: Thank you, all minor items have now been corrected.